# TpuGraphs: A Performance Prediction Dataset on Large Tensor Computational Graphs

**Phitchaya Mangpo Phothilimthana**
Google

**Sami Abu-El-Haija**
Google

**Kaidi Cao**$^*$
Stanford

**Bahare Fatemi**
Google

**Mike Burrows**
Google

**Charith Mendis**$^*$
UIUC

**Bryan Perozzi**
Google

## Abstract

Precise hardware performance models play a crucial role in code optimizations. They can assist compilers in making heuristic decisions or aid autotuners in identifying the optimal configuration for a given program. For example, the autotuner for XLA, a machine learning compiler, discovered 10–20% speedup on state-of-the-art models serving substantial production traffic at Google. Although there exist a few datasets for program performance prediction, they target small sub-programs such as basic blocks or kernels. This paper introduces TPUGRAPHS, a performance prediction dataset on full tensor programs, represented as computational graphs, running on Tensor Processing Units (TPUs). Each graph in the dataset represents the main computation of a machine learning workload, *e.g.*, a training epoch or an inference step. Each data sample contains a computational graph, a compilation configuration, and the execution time of the graph when compiled with the configuration. The graphs in the dataset are collected from open-source machine learning programs, featuring popular model architectures, *e.g.*, ResNet, EfficientNet, Mask R-CNN, and Transformer. TPUGRAPHS provides 25x more graphs than the largest graph property prediction dataset (with comparable graph sizes), and 770x larger graphs on average compared to existing performance prediction datasets on machine learning programs. This graph-level prediction task on large graphs introduces new challenges in learning, ranging from scalability, training efficiency, to model quality.

## 1 Introduction

Compilers often use performance models to solve optimization problems [28, 48], as collecting performance measurements from real hardware can be expensive, limited, or infeasible. A performance model can also be used by a compiler autotuner to evaluate candidate configurations in a search space [2, 14, 37, 53, 56]. However, developing an accurate analytical model of program performance on a modern processor is challenging and time-consuming because the underlying processor architecture, the compiler, and their interactions are complex and difficult to model analytically.

Many recent methods [2, 14, 41, 51, 66, 65, 5, 79, 45, 3, 24, 37] apply machine learning (ML) to learn performance prediction models. However, there exist only a few datasets for program performance prediction, and they all target small sub-programs. BHive [15] targets small basic blocks of assembly instructions. TenSet [81] targets ML kernels consisting of a small number of tensor operations. The database-query dataset [31] contains larger query programs, but they are still relatively small, most with fewer than 100 nodes.

---

$^*$Work partially done during internship/visiting researcher term at Google.

Unlike prior datasets, TPUGRAPHS is a performance prediction dataset on full tensor programs, represented as computational graphs. Each graph represents the main computation of an ML program, which is usually one or many training steps or one inference step. The graphs in the dataset are collected from open-source ML programs, featuring popular models (*e.g.*, ResNet, EfficientNet, Mask R-CNN, and a large variety of Transformer) for a wide range of tasks, *e.g.*, vision, NLP, speech, audio, recommendation, and generative AI. Each data sample contains a computational graph, a compilation configuration, and the execution time when executing the graph when compiled with the given configuration on a Tensor Processing Unit (TPU) v3 [39], an accelerator for ML workloads. A compilation configuration controls how the XLA compiler [70] transforms the graph for a specific optimization pass. In particular, the TPUGRAPHS dataset consists of two collections: (i) *layout* and (ii) *tile*. *Layout* configurations control how tensors are laid out in the physical memory, by specifying the dimension order of each input and output of an operation node. A *tile* configuration controls the tile size of each fused subgraph. We primarily focus on layout and tile configurations because tuning them offers the highest performance gain on average, compared to tuning other compiler optimizations.

The layout collection contains 31 million pairs of graphs and configurations, averaging over 7,700 nodes per graph. The tile collection contains 13 millions pairs of kernels and configurations, averaging 40 nodes per kernel subgraph. The layout collection is unique among existing graph datasets, in that it provides data for graph-level predictions on very large graphs. In contrast, most of the existing graph datasets fall into two categories: graph-level prediction on small graphs [11, 72, 32, 4, 32, 14, 81, 67, 83], and node-level or edge-level prediction on large graphs [29, 12, 34, 77, 86, 9, 49]. TPUGRAPHS provides 25x more graphs than MalNet [27] — the largest graph property prediction dataset with comparable graph sizes — and 770x larger graphs on average compared to TenSet [80] — the only existing large-sclae ML program performance dataset — as depicted in Figure 1. The scale of TPUGRAPHS poses several new research challenges:

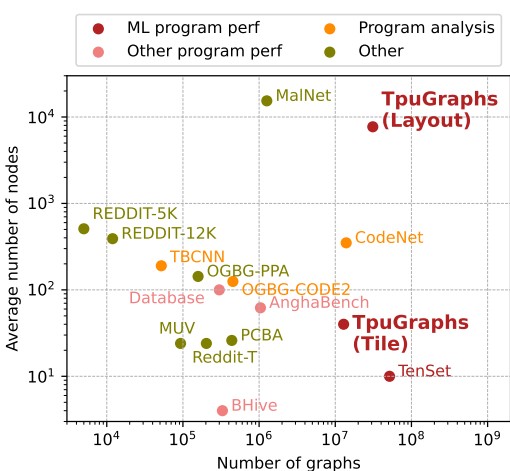

Figure 1: Scale of TPUGRAPHS compared to other graph property prediction datasets.

- How to train a neural network model that can perform graph-level predictions when the memory required to train the model on a single graph may not fit on a single device?
- How to make a model generalize well to unseen graphs when they are diverse, and the training data may be imbalanced?
- How to improve the efficiency of a training pipeline when multiple data points contain a large amount of redundant data (same core graph but different graph configurations)?

We provide baseline model code[2] based on a Graph Neural Network (GNN) [11], following the techniques from the most recent works on TPU learned cost models [41, 10]. The baseline models achieve moderate performance on both layout and tile collections. For competitive baselines, we encourage the reader to follow the Kaggle competition[3] [54].

## 2 Background & Challenges

ML compilers solve multiple optimization problems to translate an ML program, typically represented as a tensor computation graph, to an efficient executable for a hardware target. Recent works have demonstrated that search-based *autotuning* techniques can be used to generate code with close to optimal performance [13, 82, 2, 3, 71, 36, 65, 56]. However, autotuning requires a relatively large

---

[2]https://github.com/google-research-datasets/tpu_graphs
[3]https://www.kaggle.com/competitions/predict-ai-model-runtime

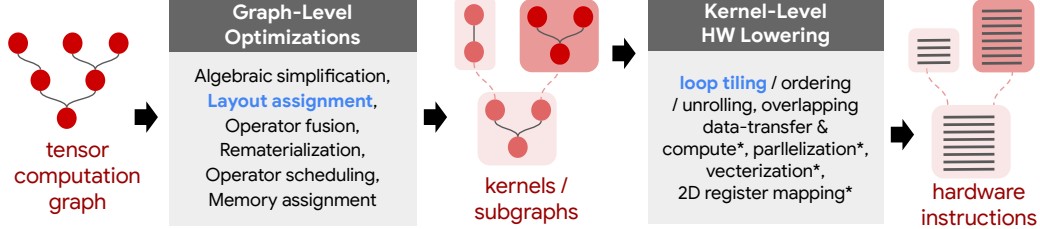

Figure 2: Important optimizations in ML compilers include graph-level and kernel-level optimizations. A graph-level optimization requires the context of the entire graph to make optimal decisions and transforms the entire graph accordingly. A kernel-level optimization transforms each kernel (a fused subgraph) at a time, independently of other kernels.

amount of resources to find quality candidates compared to traditional heuristics-based compilers. Therefore, many methods develop a learned cost model to accelerate autotuning [14, 41, 65, 79, 45, 3].

## 2.1 XLA and Autotuner

XLA [70] is a production-grade heuristics-based compiler for ML programs, capable of generating code for various hardware targets, including CPUs, GPUs, and notably TPUs [38, 39]. Figure 2 depicts important optimizations that are featured in XLA and most ML compilers. Graph-level optimizations require the context of the entire program graph to make good decisions, while kernel-level optimizations can be done independently within each kernel. A tensor computation graph is represented as High Level Operations (HLO) in XLA. Each optimization pass transforms an HLO graph into a functionally-equivalent one. The output of graph-level optimizations are a collection of kernels (represented as fused subgraphs). XLA has an accompanying autotuner [56] that can tune both graph-level and kernel-level configurations for TPUs, unlike most search-based compilers [13, 14, 79, 82, 2, 3, 71, 45, 3], which focus on kernel-level optimizations.

**Kernel-Level Optimizations.** Each node in a tensor computation graph represents a tensor operation, such as matrix multiplication, convolution, element-wise addition, *etc*. A kernel, represented as a fused subgraph, is then a fusion of multiple tensor operations. For example, Convolution-BatchNorm is a common fused kernel that appears in Convolutional Neural Networks. The most important optimization at the kernel level is tile size selection: selecting the shape of a tile of the output tensor to maximize compute efficiency of the hardware, while the required regions of input, output, and intermediate data fit in the local cache or scratchpad memory. The XLA tile size autotuner has been deployed in production to optimize the most heavily executed kernels on Google TPU fleet on a daily basis, saving approximately 2% of the total TPU compute time overall [55]. The learned cost model based on a GNN is used to select the top K most promising tile sizes to execute on real hardware [41], reducing the autotuning search time by approximately 20x.

**Graph-Level Optimizations.** At the graph level, the XLA autotuner supports tuning layout assignment, fusion, and memory space assignment passes, as well as compiler flags that control multiple optimization passes. The XLA graph-level autotuner has delivered 10–20% speedup state-of-the-art models serving substantial production traffic at Google. However, it often takes at least a few hours for the autotuner to converge when tuning one optimization pass of a single graph, and much longer for larger computation graphs. Therefore, a learned cost model would significantly reduce the search time. This motivates us to release the dataset collected from the autotuning process to advance research in developing learned performance prediction models, by addressing challenges outlined in Section 2.2, and ultimately accelerate the autotuning process for production ML workloads.

We focus on layout tuning because it offers the most speedup in general. The layout assignment pass chooses the physical layouts of the input and output tensors of each node that satisfy constraints, while minimizing program's execution time. A layout determines the order of (minor-to-major) tensor dimensions. Figure 3 displays valid input layouts in blue and the chosen layout in red. If an edge connects an output to an input with a different layout, the compiler inserts a `copy` (transpose) operator to convert the layout. In Figure 3 (left), layout of $\{1, 0, 2\}$ is assigned to the output of `add` but $\{0, 1, 2\}$

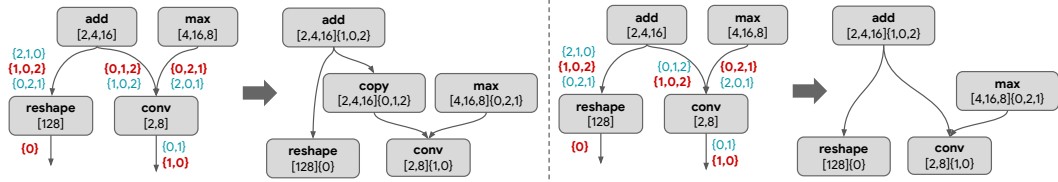

Figure 3: A node represents a tensor operator, annotated with its output tensor shape $[n_0, n_1, ...]$, where $n_i$ is the size of dimension $i$. Layout $\{d_0, d_1, ...\}$ represents minor-to-major ordering in memory. Applied configurations are highlighted in red, and other valid configurations are highlighted in blue. A layout configuration specifies the layouts of inputs and outputs of influential operators (*i.e.*, convolution, dot, and reshape). A copy operator is inserted when there is a layout mismatch.

to the first input of conv, causing a layout mismatch, unless a copy operator is inserted. The compiler must trade off between selecting the best layouts for each specific operator and the overhead of copy operators. The autotuner tunes the input-output layouts of the most layout-performance-critical nodes — *i.e.*, convolution, dot (einsum), and reshape because they are common operations and have the most constrained implementations for TPUs — and propagates layouts from these nodes to others. The autotuner picks one input-output layout combination from the valid options for each configurable node.

## 2.2 Learning Challenges

TPUGRAPHS is non-trivial because training a neural network model to make an accurate prediction on a large graph comes with multiple challenges, as follows.

**Scalability.** Existing efforts to scale GNNs have mostly focused on node-level and edge-level prediction using sampled subgraphs [29, 12, 34, 77, 86, 49, 26, 1] or graph transformations followed by local models [8, 9]. However, there is a lack of research on how to train scalable models for *property prediction of large graphs*. Training on sampled subgraphs alone is insufficient as they may not contain all the necessary information for accurate predictions. Aggregating information from the entire graph is essential for graph property prediction, but it poses challenges due to memory limits on a training device, as the memory required scales at least linearly with the size of the graph [78]. Our layout collection contains graphs of up to 44,000 nodes, so training a GNN model on an entire graph (or a batch of thereof) using a single GPU may run out of memory.

**Diversity and Imbalance of Graphs.** We want to learn a model that generalizes well to unseen graphs. However, this is non-trivial because the model must be trained on diverse types of graphs with enough samples for each type. TPUGRAPHS consists of graphs for all kinds of important ML workloads, including both inference and training, from past to present. While the dataset may be imbalanced — containing graphs from some types of architectures more than others — each graph has at least 10,000 samples of data from different configurations on average.

**Redundancy.** Another unique property of our dataset is that many samples share the same graph, which represents a large amount of redundant data. An efficient training pipeline should leverage this knowledge and reduce the redundant computation when possible. Additionally, there is another aspect of redundancy within each graph. A tensor computation graph, representing an ML workload, often consists of repeated blocks of neural network layers. The repeated blocks appear as repeated subgraphs. One may leverage this knowledge to improve the learning algorithm.

The baselines accompanying this dataset attempt to address some of these challenges, but are not close to fully solving them.

## 3 The TpuGraphs Dataset

The TPUGRAPHS dataset contains execution time data points, where each data point contains an HLO graph, its configuration, and its execution time on a single core of TPU v3. The HLO graph in

each data point is a partially optimized graph before being fed into the corresponding optimization pass. For example, in the *layout* collection, an HLO graph is the input graph to the layout assignment pass. The layout configuration of a graph is a collection of per-node layout decisions on configurable nodes (*i.e.*, convolution, dot, and reshape). For the *tile* collection, an HLO graph in each data point is a fused subgraph representing a kernel. The tile configuration of a subgraph is a configuration for the entire subgraph, not specific to any particular node.

## 3.1 Data Generation

Within our dataset, there are multiple collections of data, differing in terms of (1) the compiler optimization (*i.e.*, layout and tile), (2) the source of graphs, and (3) the search strategy.

**Graphs Collection.** We collect HLO graphs from two sources. The first source, called *XLA*, is the combination of the XLA regression benchmark — from where we collect all open-source models — and the MLPerf benchmark [50, 35]. The *XLA* graphs span diverse types of popular ML training and inference models, such as vision, NLP, speech, audio, and recommendation. The second source, called *NLP*, contains a variety of BERT for training and inference, with varying number of layers, attention heads, and hidden sizes. For each model, we run the program — written in TensorFlow, PyTorch, or JAX — and collect the largest HLO graph compiled by XLA, which represents the model's main computation. Note that a typical way that XLA handles a graph with dynamic shapes is to bucketize the graph into multiple static-shape graphs. During execution, the runtime will pad the input to match the static-shape graph with the larger closet shape. Our dataset includes graphs — for varying sequence length, batch size, model size, etc. — some of which are used for dynamic shape workloads. The TPUGRAPHS dataset is similar to the internal datasets used for prior TPU learned cost models [41, 10], but it exclusively contains graphs from open source-programs, while the internal datasets also include production models that cannot be released publicly.

**Configurations Generation.** Once we have the graphs, we use the XLA autotuner to generate data samples. The set of configurations being generated depends on how the autotuner explores the search space. For the layout collections, we ran the autotuner in two modes. The first mode explores the search space using a genetic algorithm starting from the default configuration, chosen by the compiler's heuristic. Data collected from this mode is labeled *default*. The second mode explores the search space by picking random candidates. Data collected from this mode is labeled *random*. We keep data collected in different modes in separate collections; the default collection tends to contain configurations that are not too different from the default, and have similar execution times, while the random collection includes very different configurations with very different execution times.

For the tile size tuning, the autotuner first invokes the compiler to run the graph-level optimizations and obtain fused subgraphs (kernels). For each subgraph, the autotuner enumerates all possible tile sizes for the kernel in a random order, limited by a timeout. Note that the tile size search space is much smaller than the layout search space, so we can enumerate all possible tile sizes. Therefore, there is one data collection for tile sizes. We use only the *XLA* source for graphs in this collection.

Appendix A.2 describes how we measure the execution time of a given graph and configuration.

## 3.2 Dataset Statistics and Related Datasets

Table 1 summarizes the details of the different data collections, where the collection name follows the pattern *optimization:source:search*. Table 3 in Appendix A.1 compares properties of the TPUGRAPHS dataset (all collections) against existing graph property prediction datasets.

**ML Program Performance.** The TPUGRAPHS layout collections provide more than 770x larger graphs on average compared to TenSet [80], the only existing large-scale dataset on ML program performance. Our tile collection is similar to TenSet as the configuration controls the optimization at the kernel (fused subgraph) level. However, it compliments TenSet nicely as it provides data points on different hardware. Halide Auto-scheduler [2] releases their evaluation dataset of Halide programs mainly consisting of image processing benchmarks with a few ML benchmarks.

**Other Program Performance.** Beyond ML programs, the performance prediction dataset with largest graphs is on database queries [31], whose graphs are still more than a few orders of magnitudes

Table 1: Statistics of TPUGRAPHS collections. The collection name follows the pattern *optimization:source:search*. The search may explore the same configuration multiple times, so the same pair of graph and configuration may appear multiple times with slightly different execution time from multiple measurements. The total number of samples is thus higher than the number of unique pairs.

| Collection | Core (Sub) Graphs | Avg. Nodes | Configs per Graph | Total Graphs + Configs | Samples |
|---|---|---|---|---|---|
| Layout:XLA:Default | 78 | 14,105 (372–43,615) | 10,147 (681–71,574) | **771,496** | 1,272,538 |
| Layout:XLA:Random | | | 11,648 (109–99,783) | **908,561** | 1,115,709 |
| Layout:NLP:Default | 244 | 5,659 (876–21,919) | 56,534 (9032–90,985) | **13,285,415** | 15,479,038 |
| Layout:NLP:Random | | | 66,089 (8,843–100,001) | **16,125,781** | 16,135,731 |
| Tile:XLA | 6,988 | 40 | 1,842 | **12,870,077** | 12,870,077 |

smaller than ours. Another popular performance prediction dataset is BHive [15], consisting of x86 basic blocks sourced from multiple open source programs, with runtime measurements on different Intel hardware platforms. However, the basic blocks are quite small, including four instructions on average. CompilerGym [18] releases a collection of LLVM IR code datasets that can be evaluated in their environment. The largest datasets in their collection includes AnghaBench [19] and CSmith [75]. AnghaBench provides a large number of relatively small real-world programs. CSmith programs are large (comparable to ours), but they are randomly generated programs. Additionally, CompilerGym's datasets do not come with performance measurements, so one would have to execute the programs and configurations in the CompilerGym's environment themselves to obtain program execution time.

**Program Analysis.** Other closely related datasets are on programming tasks. CodeNet [57] is a large dataset to teach AI to code, in which each code sample is a solution to one of the coding problems. OBGB-CODE2 [33] is for code summarization, containing Abstract Syntax Trees obtained from Python functions. TBCNN [52] releases its dataset on program classification from a pedagogical programming open judge system. CuBERT [40] uses Python files extracted from the ETH Py150 dataset [59] for fine-tuning and uses `github_repos` dataset under BigQuery's public-data project for pre-training. CodeBERT [25] releases its multi-programming-lingual dataset used for pre-training. Works such as inst2vec [7] and ProGraML [17] uses datasets of code in LLVM compiler intermediate representation to learn generic code representation for various program analyses and optimizations.

**Other.** Apart from code datasets, there are many other graph datasets. Open Graph Benchmark [33] suite presents graphs that are used for machine learning tasks such as GNN inference and training. GAP [6] and Graph Based Benchmark Suite (GBBS) [21] provide large-scale curated sets of graphs, primarily for evaluating traditional graph problems. SuiteSparse [43] consists of a wide variety of sparse matrices, which can be viewed as graphs. Most of these datasets are for node-level or edge-level prediction tasks. TPUGRAPHS is by far one of the largest graph property prediction datasets. TPUGRAPHS' average graph size is comparable to that of MalNet [27] — the largest scale graph property prediction dataset to date — while offering 25x more combinations of graphs and configurations. Other popular graph property prediction datasets include small molecule [58, 61], bioinformatic [22, 33], and social network datasets [62, 74].

### 3.3 Dataset Split

We split the data using 80-10-10 ratio by graphs in each collection. Splitting data by graphs ensures that graphs in the validation and test sets do not appear in the training set to evaluate the generalization of the model on unseen graphs. The validate and test graphs stay the same across different XLA collections; the same applies to NLP collections. We deliberately holdout the target labels of samples in the test set for competition purposes.

We report test and validation metrics for the tile collection by considering all configurations. For the layout collections, we report final metrics only on 1,000 configurations to reduce the computational demand for the model evaluation. We select these 1,000 configurations by sorting all configurations based on their execution times and extracting the $[0, m, 2m, \ldots, length - 1]$th configurations. The dataset includes the indices of the selected configurations[4].

---

[4] https://github.com/google-research-datasets/tpu_graphs/tree/main/tpu_graphs/evals

# 4 Learning a Performance Prediction Model

The goal of a learned cost model is to rank the performance of different configurations of a given graph. This section explains the baseline models we provide and how we train them, primarily based on the TPU learned cost model papers [41, 10].

## 4.1 Feature Extraction

TPUGRAPHS provides data in two formats: raw protobuf format and numpy arrays similar to the OGBG format [33]. The autotuner produces output results in protobuf format. A data pre-processing script converts data from the protobuf format to the numpy format. The main function of the data pre-processor is feature extraction. Node features describe the node's properties, such as output tensor shape, tensor layout, striding, padding, and operation-specific parameters. Our feature extraction is minimal. To extract a node feature vector, we either copy values from various fields in an HLO instruction (a node in an HLO graph) as they are, or convert categorical values using one-hot encoding. To convert an unbounded list of numbers (*e.g.*, tensor shape) to a fixed-size vector, we truncate the list to six elements and include the summation and/or product of all elements in the list (*e.g.*, the product of dimension sizes represents the volume of the tensor) because the tensors appearing our dataset do not contain more than six dimensions. A per-node layout configuration and tile size can be represented as a nested list with some unbounded dimensions. Similarly, we truncate these unbounded dimensions to six elements. The detailed description of node and configuration features can be found in the GitHub repo.

We provide code for training a variety of models over the numpy format. Nonetheless, the raw format can allow researchers to experiment with different feature extractions and measure impacts on the quality of a learned model.

## 4.2 Model Architecture

Figure 4 shows the model architecture we use for our baseline models, which are based on a GNN since the input program is represented as a graph. Node features consist of two parts. The first part is an opcode id, *i.e.*, type of tensor operation (such as convolution). Our baseline models map an opcode id to an opcode embedding via an embedding lookup table. The opcode embedding is then concatenated with the rest of the node features as inputs to a GNN. We combine the node embeddings produced by the GNN to create the embedding of the graph using a simple pooling reduction. The resulting graph embedding is then linearly transformed into the final scalar output by a feedforward layer. Prior work [41] has studied alternative models, including LSTM and Transformer, and shown that GNNs offer the best performance. We provide baseline models with GCN [42] and GraphSAGE [30].

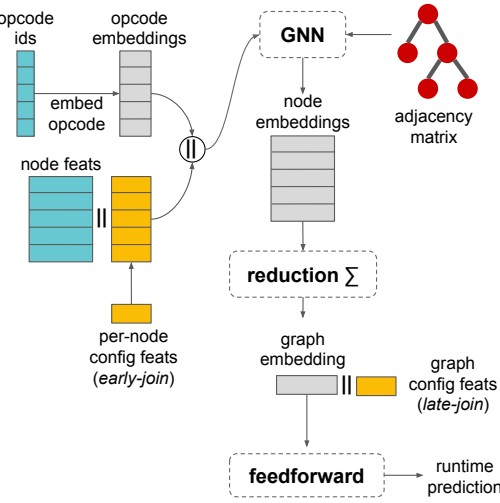

Figure 4: Model architecture.

## 4.3 Loss Functions

The primary use case of the model is to rank configurations within a given graph and select top candidates to evaluate on real hardware. Thus, we can train the model using regression losses (*e.g.*, Mean Square Error (MSE)) or ranking losses (*e.g.*, pairwise hinge loss and ListMLE [73]). A ranking loss is computed among sample pairs within the same graph in the same batch, and the losses from different graphs in the batch are reduced to get the total loss. We use Ordered Pair Accuracy (OPA) as a validation metric to select the best model checkpoint.

### 4.4 Implementation

**Layout model.** Our baseline model is a 3-layer GraphSAGE with residual connections. We concatenate node features and per-node configuration features as inputs to the GNN. If a node is non-configurable (having no layout configuration), we use a zero vector as configuration features. Our baseline code allows both a typical full graph training and a graph segment training [10]. One may improve the compute efficiency further by using historical embeddings of subgraphs and segment dropout, as in the Graph Segment Training paper.

**Tile size model.** For the tile collection, we implement three baselines: an MLP model and two GNNs (GraphSAGE and GCN with residual connections). The MLP model embeds all opcodes, concatenates with node features, sums across all nodes, and then concatenates with kernel configuration features, feeding into a 3-layer MLP. We experiment with two options to combine the graph-level kernel configuration features with the node-level information (yellow in Figure 4): either *late-join* or *early-join*. The first runs the GNN only on node features, reduces the node embeddings, and then concatenates with the graph (configuration) features. The second replicates the graph features onto every node. The early-join GraphSAGE model closely resembles the original TPU learned cost model [41].

For both models, we experiment with two objective functions: MSE and ListMLE. We find that ListMLE gives better empirical performance. Our baseline models are available at `https://github.com/google-research-datasets/tpu_graphs`. They are implemented using TensorFlow-2 and TF-GNN. The details of hyperparameters can be found in Appendix B.

## 5 Evaluation

### 5.1 Evaluation Metrics

To evaluate a model, we use two metrics. (1) Kendall's Tau assesses how well a model's ranking of configurations correlates with their corresponding runtimes. (2) *Top-K error* (or *slowdown error@K*) measures the slowdown of the chosen $K$ configurations as:

$$\text{top-K error} = \frac{\text{The best runtime of the top-}K\text{ predictions}}{\text{The best runtime of all configurations}} - 1 = \frac{\min_{i \in K} y_i}{\min_{i \in A} y_i} - 1 \qquad (1)$$

where $K$ is the top-K predictions, $A$ is all configurations of the given graph from the dataset collection, and $y$ is the measured execution time.

The choice of metrics is justified as follows. For the tile collection, since the number of configurations per graph is relatively small, one can apply the model to obtain the predictions of all configurations, choose top-K candidates according to the model to measure on real hardware, and finally select the best one according to the real measurements. On the other hand, for the layout collections, the search space is quite large. Therefore, common search strategies, such as Genetic Algorithm and Simulated Annealing, need access to a fitness function (which can be the model). Therefore, it is important that the model can well-preserve the order of the configurations (from fastest to slowest) as reflected by the correlation score.

### 5.2 Experimental Setup

**Layout model.** For each model variant, we train the model once with only a few set of hyperparameters, and select the checkpoint with the highest OPA on the validation set to evaluate its ranking correlation and top-K prediction errors. Table 5 in Appendix B reports attempted hyperparameters for modeling layout. We report the performance of the best model based on the validation score.

**Tile size model.** For each model variant, we perform hyperparameter search on opcode embedding size, hidden size, network depth, and learning rate, considering values specified in Table 5. Unlike in the layout collections where the accuracy of each model is quite stable across multiple training runs, the accuracy of a model in the tile collection fluctuates dramatically. Therefore, we train each model variant three times. For each run, we select the model checkpoint with the highest OPA on the validation set. We report the top-K errors of the run that achieves the median top-1 error on

Table 2: Kendall's Tau correlation and prediction errors (Eq. 1) of our best baseline model on different dataset collections. The values of $(K_1, K_2, K_3)$ are (1, 10, 100) for the layout collections, and (1, 5, 10) for the tile collection.

| Collection | Kendall $\tau$ | | Top-$K_1$ E % | | Top-$K_2$ E % | | Top-$K_3$ E % | |
|---|---|---|---|---|---|---|---|---|
| | Val | Test | Val | Test | Val | Test | Val | Test |
| Layout:XLA:Random | 0.19 | 0.34 | 19.8 | 10.9 | 12.3 | 5.7 | 9.7 | 1.6 |
| Layout:XLA:Default | 0.12 | 0.21 | 3.8 | 14.1 | 1.9 | 0.6 | 0.3 | 0.2 |
| Layout:NLP:Random | 0.58 | 0.53 | 2.1 | 4.6 | 2.0 | 1.0 | 0.6 | 0.09 |
| Layout:NLP:Default | 0.30 | 0.28 | 4.0 | 4.0 | 3.7 | 3.1 | 3.5 | 0.13 |
| Tile:XLA | – | – | 10.5 | 9.1 | 3.0 | 4.2 | 1.8 | 2.8 |

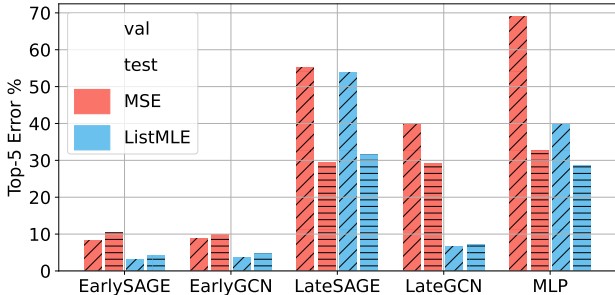

Figure 5: Prediction errors (%) of different model variants on the Tile:XLA collection. *Early* and *Late* refer to *early-join* and *late-join* options.

the validation set. An aggregated top-K error is an average across all kernels in the graphs in the validation/test set. Note that the number of kernels varies across graphs.

## 5.3 Results on Layout Collections

Table 2 reports the top-K slowdown errors and Kendall's Tau correlation of the best model across all graphs (programs) in the validation and test sets for each dataset collection. According to the correlation scores, the layout collections on the default search space are more difficult than those on the random search space. This result matches our intuition because the default search space contains many similar configurations near the default, so it is difficult to rank them; whereas, the random search space contains more diverse configurations. The XLA layout collections are also noticeably more difficult than the NLP layout collections. This is also expected because the XLA collections contain more diverse graphs, while the NLP collections contain only graphs with the Transformer architecture. In terms of top-K errors, the model struggles to identify fast candidates on Layout:XLA:Random. If we use the learned cost model to select the top configuration, we will be on average 10–20% slower than the known optimal configuration. Even if we consider the top 10 candidates, we will still be on average 5–13% off. We hypothesize that this is due to the combination of the diversity of both graphs and configurations in this collection. The correlation and top-K errors vary wildly across graphs (programs) as shown in Table 7 in Appendix C.

## 5.4 Results on Tile Collection

Table 2 also reports the average top-K errors on the tile collection. The average top-1 error of the best model is comparable to the original TPU learned cost model paper's [41]. Figure 5 compares alternative choices. Similar to the original paper, our results show that combining configuration features with node features early (*early-join*) is superior to combining configuration features with a reduced graph embedding later (*late-join*). Using a ranking loss (ListMLE) is much more effective than using MSE. Additionally, we compare the choice of a GNN between GraphSAGE and GCN, and find they are comparable. We also provide an MLP baseline without a GNN, and confirm that a GNN is essential to achieve good accuracy.

Appendix C reports additional results including per-graph evaluation metrics, an additional ablation study, and model's prediction overhead. Note that our dataset is not exactly the same as the internal dataset used in the original papers [41, 10], but they share a large number of overlapping graphs.

# 6 Discussion and Future Directions

There are many potential improvements to be made on top of the baseline models we provide. First, we observe that a tensor computation graph typically contains repeated subgraphs, representing repeated blocks of neural network layers. One direction is to leverage this repeated structure to devise a more compact representation that is easier to learn. Second, as mentioned earlier, the dataset may contain some types of graphs, *e.g.*, ResNet, significantly more than others. This skew may make the learned model perform well on common types of graphs, but poorly on uncommon types. One may investigate how to address this data imbalance problem to improve the quality of the learned model. Finally, while we know that developing a purely analytical cost model is extremely difficult, training an accurate learned cost model is not easy either, especially when graphs are large. One idea is to combine the best of both worlds, using analytical modeling when easy to do and letting the learned model make corrections to the analytical estimates.

We plan to continue improving our dataset in multiple aspects. First, we would like to include more diverse graphs. Prior approaches generate random programs for training data [2, 14, 5]. We deliberately avoid randomly generated programs in our dataset because we would like a model trained on the dataset to achieve high performance on realistic programs used in production, instead of achieving moderate performance on both real-word and randomly generated programs. However, we acknowledge that the diversity of graphs in the dataset is extremely important for the generalization of the model. One way to generate more realistic tensor programs is to leverage Neural Architecture Search [84, 68, 69, 85, 47, 46, 60, 64, 76, 23]. We leave this as future work, potentially the next version of the dataset. Second, we would like to include data measured on other hardware platforms beyond TPUs, such as CPUs and GPUs. Nonetheless, we believe that the general techniques of training an accurate learned performance model (*e.g.*, improvements on GNNs, Graph Segment Training method, etc.) are applicable to other hardware targets; therefore, the improvements coming out from experimenting with the current version of the dataset should also benefit other hardware platforms as well. Many compiler optimizations are also common across multiple hardware backends. For example, the tile size selection has shown to be one of the most important optimizations across all widely used hardware (*i.e.*, CPUs, GPUs, and TPUs) and even custom accelerators. Layout optimizations are also applicable on CPUs and GPUs, but the layout options on CPUs and GPUs may be limited if the compiler depends on pre-optimized library kernels.

We hope that TPUGRAPHS will propel advances in compilers. In particular, researchers may be able to extract insights on how to improve code generation for tensor programs. For example, which information in a tensor computation graph is important to make various optimization decisions? How can we build an accurate cost model for an important class of hardware architectures? Can a learned representation for tensor programs guide various tensor compiler optimizations?

## Acknowledgement

We would like to thank Sai Ganesh for a pointer to generate open-source XLA HLO graphs, Mark Daoust for help on the dataset's hosting location, and Amit Sabne and David Majnemer for reviewing the dataset's security information. Charith's contributions were partially supported by ACE, one of the seven centers in JUMP 2.0, a Semiconductor Research Corporation (SRC) program sponsored by DARPA and NSF under grant CCF-2316233.

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

# A  Additional Dataset Information

**Documentation.** The documentation of the dataset can be found at `https://github.com/google-research-datasets/tpu_graphs`. The github repo contains instructions and code on how to download and use the dataset.

**License.** The dataset is licensed under the Creative Commons Attribution 4.0 International License (CC-BY). To view a copy of this license, visit `http://creativecommons.org/licenses/by/4.0/`. All code is licensed under the Apache License, Version 2.0 (Apache 2.0); You may obtain a copy of the Apache 2.0 license at: `https://www.apache.org/licenses/LICENSE-2.0`.

**Author Statement.** The authors bear all responsibility in case of violation of rights. The authors will monitor the issues and provide necessary maintenance to ensure the access to the data.

## A.1  Dataset Comparison

Table 3: Comparison of TPUGRAPHS properties with other large-scale graph property prediction datasets. * provide only programs, but one may use them in CompilerGym [18] environment to obtain performance measurements when compiling with specific configurations. † provides randomly generated programs.

| Application | Dataset | Graphs (+ Configs) | Avg. Nodes |
|---|---|---|---|
| ML Program Perf | TPUGRAPHS (Layout) | 31,091,253 | 7,705 |
| | TPUGRAPHS (Tile) | 12,870,077 | 40 |
| | TenSet [80] | 51,577,248 | 5–10 |
| Other Program Perf | Database [31] | 300,000 | < 100 |
| | BHive [15] | 330,018 | 4 |
| | AnghaBench* [19] | 1,041,333 | 62 |
| | CSmith*† [75] | 530,000 | 5,845 |
| Program Analysis | CodeNet [25] | 13,916,868 | 200–500 |
| | OGBG-CODE2 [33] | 452,741 | 125 |
| | TBCNN [52] | 52,000 | 190 |
| Cybersecurity | MalNet [27] | 1,262,024 | 15,378 |
| Molecule | PCBA [58] | 437,929 | 26 |
| | MUV [61] | 93,087 | 24 |
| Bioinfomatic | DD [22] | 1,178 | 284 |
| | OGBG-PPA [33] | 158,100 | 243 |
| Social Network | Reddit-T [62] | 203,088 | 24 |
| | REDDIT-12K [74] | 11,929 | 391 |
| | REDDIT-5K [74] | 4,999 | 509 |

## A.2  Execution Time Measurement

We measure the execution of a compiled binary on a *single TPU chip* using *random input data*. Note that some of the graphs in the layout collection must be run on multiple TPU chips. However, doing so is not economically viable for autotuning a large number of models and generating the dataset. Therefore, the autotuner modifies the final optimized graph (after all graph-level optimizations) to make it runnable on a single TPU chip in two ways. First, we replace each collective communication operation (*e.g.*, all-reduce and all-gather) with a no-op that simply allocates the right amount of output buffer (with undefined values). This means the measured execution time ignores the time taken by collective operations. We think this is reasonable because layout decisions rarely affect the execution time of collective operations, and we care about the ranking of execution time rather than the absolute time. The second modification we perform is to replace dynamic loop bounds with fixed loop bounds. Without such replacement, a dynamic loop bound may depend on random input data, resulting in an extremely large loop bound, making the program run unrealistically slowly. The use of random or undefined data, however, does not affect the execution time of a compute operation (*e.g.*, convolution) because the timing does not depend on the input data. Because of these modifications, our absolute execution time measurement may be inaccurate in some cases, but this estimation approach has been used in production to tune graph-level optimizations and deliver large speedups on many important

models. Therefore, we believe it is reasonable to use the execution time measured by the approach outlined here as a prediction target.

### A.3 Graphs in Dataset

**XLA Collection.** Graphs in the XLA collection are collected from open-source models from the following sources:

- `https://github.com/tensorflow/models`
- `https://github.com/tensorflow/tensorflow`
- `https://github.com/tensorflow/tensor2tensor`
- `https://github.com/tensorflow/tpu`
- `https://github.com/google/brax`
- MLPerf [50, 35]

**NLP Collection.** Graphs in the NLP collection are all collected from TensorFlow Hub. Table 4 reports the architectures and hyperparameters of the models used to generate graphs in this collection.

Table 4: Architectures and hyperparameters of models in the NLP collection.

| Model Name | Source | Layers | Hidden Size | Attention Heads |
|---|---|---|---|---|
| bert_en_uncased | Devlin et al. [20] | 12, 24 | 768, 1024 | 12, 16 |
| bert_en_wwm_uncased | Devlin et al. [20] | 24 | 1024 | 16 |
| bert_en_cased | Devlin et al. [20] | 12, 24 | 768, 1024 | 12, 16 |
| bert_en_wwm_cased | Devlin et al. [20] | 24 | 1024 | 16 |
| bert_multi_cased | Devlin et al. [20] | 12 | 768 | 12 |
| small_bert | Devlin et al. [20] | 2, 4, 6, 8, 10, 12 | 128, 256, 512, 768 | 2, 4, 8, 12 |
| albert_en | Lan et al. [44] | 12, 24 | 768, 1024, 2048, 4096 | 12, 16, 32, 64 |
| electra | Clark et al. [16] | 12, 24 | 256, 768, 1024 | 4, 12, 16 |
| experts_pubmed | TensorFlow Hub | 12, 24 | 768, 1024 | 12, 16 |
| experts_wiki_books | TensorFlow Hub | 12, 24 | 768, 1024 | 12, 16 |
| talking_heads | Shazeer et al. [63] | 12, 24 | 768, 1024 | 12, 16 |

## B  Hyperparameters

Table 5: Hyperparameters of our baseline models. We report the result of the best version of each baseline variant by tuning opcode embedding size, hidden size, and number of GNN layers from the values specified.

| Parameter | Considered Values | |
|---|---|---|
| | Layout Models | Tile Size Models |
| Opcode embedding size | 64, 128 | 64, 128, 256 |
| Hidden size | 100, 200 | 64, 128 |
| GNN layers | 2, 3, 4 | 2, 3 |
| Batch size | 20 | 100 |
| Learning rate | 0.001 | 0.01, 0.001 |
| Training iterations | 200 | 500 |
| Loss | MSE, ListMLE | MSE, ListMLE |
| Optimizer | Adam | Adam |

## C  Additional Results

### C.1 Prediction Accuracy

**Layout Collections.** Table 6 compares alternative choices in terms of the model architecture and the training method on the Layout:XLA:Random collection. **GST** implements the Graph Segment Training method [10] without historical embedding and segment dropout. Unlike in the original paper, we partition a graph based on a topological order of nodes. At training time, we randomly pick a segment — covering nodes $\in [i, i + \text{segment length})$, where $i$ is drawn at random — for backward

Table 6: Kendall's Tau correlation and prediction errors (%) of different model variants and training methods on the Layout:XLA:Random collection.

| Model | Kendall $\tau$ | | Top-1 E % | | Top-10 E % | | Top-100 E % | |
|---|---|---|---|---|---|---|---|---|
| | Val | Test | Val | Test | Val | Test | Val | Test |
| GST | 0.13 | 0.32 | 29.2 | 8.1 | 12.6 | 6.2 | 6.0 | 2.3 |
| Full Graph | 0.19 | 0.37 | 33.3 | 12.2 | 14.5 | 4.8 | 6.8 | 3.3 |
| MSE loss | 0.03 | 0.31 | 24.5 | 8.1 | 17.7 | 5.4 | 6.0 | 1.3 |
| Random | 0.002 | 0.007 | 24.3 | 97.1 | 4.7 | 10.0 | 0.1 | 1.9 |

Table 7: Per-program Kendall's Tau correlation and prediction errors (%) on the validation set of the Layout:XLA collections of a few selected models.

| | Layout:XLA:Random | | | | | | Layout:XLA:Default | |
|---|---|---|---|---|---|---|---|---|
| Program | GST | | Full Graph | | MSE loss | | GST | |
| | Kendall | Top-1 | Kendall | Top-1 | Kendall | Top-1 | Kendall | Top-1 |
| `bert_pretraining.4x4.fp16` | 0.23 | 8.9 | 0.65 | 0.0 | 0.04 | 3.2 | 0.25 | 8.9 |
| `inception_v3_batch_128_train` | -0.34 | 70.3 | -0.49 | 63.8 | -0.48 | 63.4 | 0.27 | 3.0 |
| `mlperf_bert_batch_24_2x2` | 0.39 | 0.5 | 0.24 | 0.5 | -0.01 | 21.1 | 0.21 | 7.2 |
| `resnet50.4x4.fp16` | 0.13 | 53.6 | 0.10 | 69.0 | 0.12 | 53.6 | 0.15 | 9.6 |
| `resnet_v1_50_official_batch_128_bf16` | 0.10 | 17.1 | 0.07 | 17.1 | 0.09 | 66.6 | -0.05 | 0.7 |
| `tf2_bert_pretrain_dynamic_batch_size` | 0.52 | 5.5 | 0.66 | 5.9 | 0.01 | 19.5 | 0.18 | 1.4 |
| `unet_3d.4x4.bf16` | 0.23 | 1.6 | 0.19 | 1.6 | 0.30 | 1.6 | -0.03 | 0.0 |

propagation. Note that all nodes and edges are used in a forward pass. We use a segment length of 5,000. Results in the main paper (Table 2) are from the GST model. **Full Graph** uses the entire graph for forward and backward passes during training (a typical method). Unlike all other models, **MSE loss** uses Mean Squared Error loss function: it aims to model the exact runtime of every configuration, rather than the ranking of configurations.

Table 7 further reports per-program ranking correlation and top-K errors of a few selected baseline models on the programs (graphs) in the validation set. As shown in the table, the evaluation scores vary widely across different graphs. We show the scores only on the validation set because we do not want to reveal the details about the test set for competition purposes.

**Tile Collection.**    Table 8 summarizes the overall average prediction errors of all the baseline models. Table 9 further reports per-program average prediction errors across each program's kernel subgraphs of a few selected model variants with their best hyperparameter values.

## C.2    Prediction Overhead

Table 10 reports the evaluation time of a configuration when using the real evaluation (compiling using a CPU and executing on a TPU) and using the model prediction (on a CPU) for graphs in the validation set. The real evaluation takes 94–2400x longer than the model prediction. This confirms that it is significantly cheaper to run a learned cost model to estimate the execution time, than to measure the actual execution time (which would require a long compilation). Also note that the graph feature extraction time can be amortized across multiple configurations of the same graph, so the model prediction will be even faster in practice.

Table 8: Prediction errors (%) on the Tile:XLA collection of different models, where *Early* and *Late* refer to the *early-join* and *late-join* options of the configuration features. Each line trains three times (with the best hyperparameter configuration per architecture), and chooses the model with the median performance (according to the top-1 error on the entire validation set).

| Model / Training | | Top-1 Error % | | Top-5 Error % | | Top-10 Error % | |
|---|---|---|---|---|---|---|---|
| Loss | Type | Val | Test | Val | Test | Val | Test |
| ListMLE | EarlySAGE | 10.5 | 9.1 | 3.0 | 4.2 | 1.8 | 2.8 |
| | EarlyGCN | 11.5 | 11.4 | 3.6 | 4.7 | 2.4 | 3.3 |
| | LateSAGE | 124.8 | 71.8 | 53.8 | 31.7 | 30.3 | 19.2 |
| | LateGCN | 17.2 | 17.4 | 6.7 | 7.3 | 4.4 | 4.9 |
| | MLP | 84.5 | 52.9 | 39.7 | 28.7 | 25.6 | 17.0 |
| MSE | EarlySAGE | 19.0 | 22.0 | 8.2 | 10.3 | 5.6 | 6.5 |
| | EarlyGCN | 19.4 | 21.0 | 8.7 | 9.9 | 6.4 | 7.1 |
| | LateSAGE | 117.0 | 67.4 | 55.2 | 29.5 | 33.4 | 18.6 |
| | LateGCN | 87.7 | 67.0 | 40.0 | 29.2 | 27.3 | 20.6 |
| | MLP | 117.5 | 65.3 | 69.0 | 32.8 | 48.7 | 22.3 |

Table 9: Per-program prediction errors (%) on the validation set of the Tile:XLA. Each line trains three times (with the best hyperparameter configuration per architecture), and chooses the model with the median performance (according to the top-1 error on the entire validation set). These models are trained with ListMLE. A top-K error of a program is an average of top-K errors of all kernels in the program. The number of kernels per program is in paranethesis.

| Program (number of kernels) | MLP | | EarlyGCN | | EarlySAGE | | LateGCN | | LateSAGE | |
|---|---|---|---|---|---|---|---|---|---|---|
| Error at top: | 1 | 5 | 1 | 5 | 1 | 5 | 1 | 5 | 1 | 5 |
| `bert_pretraining.4x4.fp16 (56)` | 60 | 25 | 7 | 2 | 6 | 1 | 19 | 5 | 118 | 43 |
| `inception_v3_batch_128_train (264)` | 119 | 53 | 8 | 2 | 6 | 1 | 10 | 3 | 170 | 77 |
| `mlperf_bert_batch_24_2x2 (75)` | 34 | 14 | 13 | 5 | 15 | 4 | 23 | 10 | 105 | 33 |
| `resnet50.4x4.fp16 (103)` | 78 | 33 | 16 | 3 | 15 | 4 | 21 | 8 | 97 | 33 |
| `resnet_v1_50_official_batch_128_bf16 (108)` | 75 | 49 | 15 | 6 | 13 | 4 | 23 | 11 | 91 | 50 |
| `tf2_bert_pretrain_dynamic_batch_size (60)` | 37 | 13 | 8 | 2 | 5 | 2 | 18 | 7 | 60 | 19 |
| `unet_3d.4x4.bf16 (10)` | 110 | 53 | 30 | 3 | 48 | 6 | 17 | 0 | 124 | 69 |

Table 10: Real evaluation time vs model prediction time. The 'Batch Inference' column reports time to perform batch inference on 100 configurations divided by 100, while the 'Inference' column reports time to perform inference on a single configuration. The real evaluation takes 94–2400x longer than the model prediction.

| Program | Real Evaluation Time (s) | | Model Prediction Time (s) | | |
|---|---|---|---|---|---|
| | Compilation | Execution | Feature Extraction | Batch Inference | Inference |
| `bert_pretraining.4x4.fp16` | 45 | 2.8 | 0.2 | 0.02 | 0.1 |
| `inception_v3_batch_128_train` | 135 | 3.6 | 0.3 | 0.01 | 0.09 |
| `mlperf_bert_batch_24_2x2` | 104 | 22 | 0.6 | 0.02 | 0.1 |
| `resnet50.4x4.fp16` | 126 | 0.9 | 0.2 | 0.006 | 0.08 |
| `resnet_v1_50_official_batch_128_bf16` | 99 | 1.7 | 0.1 | 0.07 | 0.08 |
| `tf2_bert_pretrain_dynamic_batch_size` | 44 | 2.8 | 0.4 | 0.02 | 0.1 |
| `unet_3d.4x4.bf16` | 475 | 1.4 | 0.1 | 0.004 | 0.08 |

