# OpenReview forum: "TpuGraphs: A Performance Prediction Dataset on Large Tensor Computational Graphs"
_NeurIPS.cc/2023/Track/Datasets_and_Benchmarks — NeurIPS 2023 Datasets and Benchmarks Poster_

### Official Review · Reviewer_2Nc2 · 2023-07-20
**Made large dataset for cost model training**

**Rating:** 5
**Confidence:** 4
**Clarity:** Yes, this paper is well written

**Strengths:**

1) This paper has a good contribution in that it discloses the actual dataset and shows the meaningful experimental results using it.
2) The fact that GNN-based layout, tiling optimization was suggested in the neural network compilation stage also has great significance.

**Additional Feedback:**

No more additional feedbak.

**Correctness:**

In general, GA is an approach based on randomness. How can you guarantee that the Default method based on GA could be distinguished from Random method?

**Documentation:**

1) This paper describes the details of the dataset
2) However, download link for the dataset is currently unavailable

**Ethics:**

Irrelevant

**Limitations:**

The proposed model takes compiler configurations as one of input. It seems that the learned model as well as dataset are applicable only to the specific compiler considered in this paper.

**Opportunities For Improvement:**

1) It would be nice if the author could explain why do we need different models (different parameters) for different optimization domain (e.g., tile size, layout)
2) It would be better if the author emphasizes how important and difficult the layout optimization problem is.

**Relation To Prior Work:**

1) GST[8] is well explained how this paper uses the GST to enable multi-node-level training
2) Model architecture defined in this paper looks the same as the model architecture defined in the [35] work. It would be nice to add a section to compare the current model with the previous one.
3) Tile size selection is already addressed in the previous work[35]

**Summary And Contributions:**

1) This paper proposed dataset that can be used for cost model training, and it also designed AI model that can be trained not only to predict TPU performance for various type of DL applications (e.g., CNN, transformer) but also to make optimal decision
2) Made large dataset for cost model training publicly available
3) Used the GST method to enable training many nodes together, which can help enhance inference accuracy
4) Verified its effectiveness using well-known optimization methods (layout, tile-size selection) widely used in DL compilers

---

> ### Author Response · Authors · 2023-08-17
>
> We thank the reviewer for the valuable feedback and insightful comments. We appreciate the reviewer’s acknowledgement of the usefulness of our dataset. We hope we are able to address the reviewer’s concerns, and respectfully ask the reviewer to consider increasing the score.
>
> #### **Different models for different tasks**
> We know that the layout optimization and the tile size optimization are quite different because the former is applied at the graph level, but the latter is applied at the kernel subgraph level. Therefore, we think it is more appropriate to train different models for these two tasks. However, it would be an interesting experiment to run and validate if a model trained on both tasks would perform better or worse than separate models. Users of our dataset should be able to conduct such an experiment.
>
> #### **Importance and difficulty of layout optimization**
> The paper on the XLA autotuner [46] describes the layout optimization in detail. We will refer the readers to [46] and clarify how big the search space is (exponential in terms of the number of convolution and reshape operations in the graph).
>
> [46] Phitchaya Mangpo Phothilimthana et al. A Flexible Approach to Autotuning Multi-Pass Machine Learning Compilers. PACT, 2021.
>
> #### **Baseline models**
> Our intention is to provide the best model we use in production internally [35] as the baseline model for our dataset. However, converting [35] to open source is non-trivial, so we decided to reimplement [35] using TF-GNN; therefore, the open source baseline model and [35] should be very similar mathematically. We will clarify this in the paper.
>
> #### **Incorrect dataset URL**
> We apologize that the dataset URL provided in the main paper is incorrect. The dataset approval process — which had not been finalized when we submitted the main paper — enforces us to follow a certain naming convention, so the correct URL is https://github.com/google-research-datasets/tpu_graphs, as specified on the submission page and in the appendix.

---

> > ### Comment · Reviewer_2Nc2 · 2023-08-25
> >
> > Thank the authors for the detailed answers. Many of the concerns have been resolved.
> > Nevertheless, I still have the following concern.
> >
> > "The proposed model takes compiler configurations as one of input. It seems that the learned model as well as dataset are applicable only to the specific compiler considered in this paper."
> > > It seems that the author of this paper didn't address this comment. This reviewer wonders if the proposed approach is only applicable to the XLA compiler for TPU HW.
> >
> > I appreciate again the thorough response.

---

> > > ### Author Response · Authors · 2023-08-25
> > >
> > > Sorry for missing one of your comments!
> > >
> > > Yes, the current dataset contains data collected from programs compiled by the XLA compiler running on TPUs. We would like to point out that collecting compiler configurations and runtimes for different hardware backends or different compilers requires significant changes to our infrastructure, so we cannot easily construct the dataset for other hardware devices and compilers. However, we believe that the general techniques of training an accurate learned performance model (e.g. improvement on GNNs, Graph Segment Training method, etc) are applicable to other hardware targets and compilers, so the improvements coming out from our dataset should also benefit other hardware platforms as well. Many optimizations are common across different compilers and hardware backends. For example, the tile size selection has shown to be one of the most important optimizations across all compilers (e.g., TVM, Ansor, Halide, etc) and all widely used hardware (i.e., CPUs, GPUs, and TPUs) and even custom accelerators. Layout optimizations are also applicable on CPUs and GPUs, but the layout options on CPUs and GPUs are quite limited if the compiler depends on pre-optimized library kernels (e.g., CuDNN and MKL). We will consider including data for other hardware and compilers in future releases of the dataset.

---

### Official Review · Reviewer_jdZ8 · 2023-07-21
**Large-scale dataset for TPU Performance Autotuning**

**Rating:** 7
**Confidence:** 4

**Strengths:**

The paper demonstrates its robustness by addressing several formidable challenges associated with learning on large graphs through the novel TpuGraphs dataset.
Scalability poses a significant hurdle in training neural network models to make accurate predictions on large graphs. While existing efforts have concentrated on node-level and edge-level prediction using sampled subgraphs, devising scalable models for predicting properties of large graphs has remained an understudied area. Aggregating information from the entire graph is crucial for accurate graph property prediction, but it is hampered by the memory constraints on training devices, as the memory requirements scale linearly with the graph's size. TpuGraphs fills this gap by presenting graphs with up to 44,000 nodes, presenting an ambitious endeavor in the field of graph property prediction.

Ensuring generalizability of the learned model necessitates exposure to diverse graph types with sufficient samples for each type. TpuGraphs addresses this concern comprehensively by encompassing a wide range of machine learning workloads, encompassing both inference and training scenarios across various time periods, thereby enhancing the model's ability to generalize. Although some graphs may exhibit an imbalance in their representation across different architectures, each graph comprises an ample number of samples (averaging at least 10,000 samples) from distinct configurations, resulting in a robust learning process.

The unique characteristic of TpuGraphs lies in multiple samples sharing the same graph, leading to the presence of redundant data. Leveraging this redundancy efficiently during the training process is of paramount importance to minimize computational overhead. Additionally, recognizing repeated blocks of neural network layers within each graph, appearing as repeated subgraphs, presents an avenue for further optimizing the learning algorithm. While the paper acknowledges that the baselines accompanying the dataset attempt to tackle some of these challenges, they are yet to fully conquer them, indicating promising prospects for future research.

**Additional Feedback:**

N/A

**Clarity:**

The paper is exceptionally well written. The authors have skillfully highlighted the unique strengths of TPUGRAPHS, such as its extensive size and representation of popular model architectures, making it a valuable resource for performance prediction in machine learning workloads.

**Correctness:**

The paper is sound and correct. The evaluation metrics are fair and appropriate.

**Documentation:**

The documentation is clear, complete and easy to follow.

**Limitations:**

One limitation of the paper is the lack of mention of dynamic shape support in the TPUGRAPHS dataset. Dynamic shapes, where the size of tensors varies during computation, are prevalent in modern machine learning models, especially in natural language processing tasks. However, the paper does not explicitly address how the dataset handles graphs with dynamically shaped tensors. This limitation might restrict the dataset's applicability to certain models or scenarios that heavily rely on dynamic shapes, leading to potential difficulties or inaccuracies in predictions for such cases.

The focus of TPUGRAPHS on Tensor Processing Units (TPUs) is another limitation. While TPUs are powerful hardware accelerators designed for machine learning workloads, the dataset overlooks the performance prediction aspect on other widely used hardware platforms, such as GPUs and CPUs. Ignoring GPUs and CPUs might limit the dataset's usefulness for optimizing performance across a broader range of hardware configurations, hindering its potential applicability in heterogeneous computing environments.

Model diversity. The paper mentions that TPUGRAPHS contains graphs from popular model architectures like ResNet, EfficientNet, Mask R-CNN, and Transformer, including BERT models. However, there might be a question about whether those workloads are diverse enough to cover the entire spectrum of models used in practical applications.

Addressing these limitations would not only enhance the value and applicability of the TPUGRAPHS dataset but also strengthen the overall impact of the research in the field of performance prediction and optimization for machine learning workloads. Future work could consider incorporating support for dynamic shapes, extending the hardware scope to include GPUs and CPUs, and broadening the diversity of more models to ensure more comprehensive and robust performance predictions.

**Opportunities For Improvement:**

There are several opportunities for improvement in the context of the TPUGRAPHS dataset and the research surrounding performance prediction for large graphs in machine learning. Some potential opportunities include:

**Dynamic Shape Support.** Addressing the limitation of dynamic shape support would be a significant opportunity for improvement. Incorporating graphs with dynamically shaped tensors into the dataset and developing models that can handle variable-sized inputs would enhance the dataset's applicability to a broader range of machine learning models, especially in tasks such as natural language processing and object detection.

**Hardware Diversity.** Expanding the hardware scope beyond TPUs to include GPUs and CPUs would provide more comprehensive insights into performance prediction and optimization across different hardware platforms. Researchers could investigate how machine learning workloads perform on various hardware architectures, enabling better-informed decisions on hardware selection and optimization.

**Model Architectures.** Continuously updating and expanding the variety of model architectures in the TPUGRAPHS dataset would be beneficial. Including a more diverse set of model architectures, variations of existing models, and emerging models would provide researchers with a more representative dataset for performance prediction across a wider range of machine learning applications.

**Benchmarking and Comparison.** Establishing standardized benchmarks and evaluation metrics for performance prediction models would enable fair comparisons between different approaches. This would facilitate advancements in the field and foster collaboration among researchers.

**Graph Pruning and Compression.** Exploring techniques to prune or compress large graphs while preserving critical information for performance prediction could lead to more efficient training and deployment of models on resource-constrained devices.

**Real-time Performance Prediction.** Developing methods for real-time performance prediction during model execution could enable dynamic runtime optimizations, leading to better adaptability of machine learning workloads to changing hardware conditions and varying resource availability.

Overall, these opportunities for improvement present exciting avenues for research and innovation in the field of performance prediction and optimization for machine learning workloads on large graphs. Addressing these areas could lead to more robust and efficient machine learning systems, ultimately benefiting a wide range of applications and industries.

**Relation To Prior Work:**

The paper provides a comprehensive and insightful discussion of its relationship with previous research. The authors skillfully navigate through the existing body of work, offering valuable insights into how their study builds upon and extends prior investigations in the field. This thoughtful analysis of the literature enhances the paper's credibility and contributes to a well-rounded understanding of the research landscape.

**Summary And Contributions:**

This paper introduces TpuGraphs, a largest-scale dataset so far, for performance auto tuning. The dataset is 25 times larger than the existing one, and the graph in each datapoint is 770 times larger on average. It covers workloads extracted from all major neural network architectures in vision, NLP and audio. The authors highlight the potential applications of this dataset, such as assisting autotuners in achieving significant speedup on state-of-the-art models, and aiding in the optimization of machine learning programs running on TPUs.

---

> ### Author Response · Authors · 2023-08-17
>
> We thank the reviewer for the valuable feedback and insightful comments. We appreciate the reviewer for confirming that our dataset is unique and technically challenging.
>
> #### **Dynamic shape**
> A typical way that XLA handles a graph with dynamic shapes is to bucketize the graph into multiple static-shape graphs. During execution, the runtime will pad the input to match the static-shape graph with the larger closet shape. This is the method used for the majority of ML workloads at Google for both training and inference. Our dataset includes graphs for varying sequence length, batch size, model size, etc, some of which are used for dynamic shape workloads. We will clarify this in the paper.
>
> #### **Other hardware platforms**
> Collecting compiler configurations and runtimes for different hardware backends requires significant changes to our infrastructure, so we cannot easily construct the dataset for other hardware devices. However, we believe that the general techniques of training an accurate *learned performance model* (e.g. improvement on GNNs, Graph Segment Training method, etc) are applicable to other hardware targets, so the improvements coming out from our dataset should also benefit other hardware platforms as well. Many compiler optimizations are also common across multiple hardware backends. For example, the tile size selection has shown to be one of the most important optimizations across all widely used hardware (i.e., CPUs, GPUs, and TPUs) and even custom accelerators. Layout optimizations are also applicable on CPUs and GPUs, but the layout options on CPUs and GPUs are quite limited if the compiler depends on pre-optimized library kernels (e.g., CuDNN and MKL). We will consider including data for other hardware in future releases of the dataset.
>
> #### **Model architectures**
> Our dataset already includes variations of existing models (e.g. varying sequence length, batch size, model size, etc) to some certain extent, with more coverage for NLP graphs. However, the reviewer’s suggestion on continuously updating and expanding the variety of model architectures is an excellent idea, and we will try our best to keep releasing newer versions of the dataset to include more models.
>
> #### **Benchmarking and comparison**
> We totally agree with the reviewer that establishing standardized benchmarks and evaluation metrics on the dataset is important to enable fair comparisons. In fact, we proposed a competition for NeurIPS’23 workshop, and the proposal was accepted. We intend to launch the competition at the end of this month. Tentatively, the **final model score** will be the average error (equation 5) on all 5 collections, and we may reward the top-performing team on each metric. We will include more details in the final version of the paper. We also wish to use the findings from the competition to help refine the evaluation metrics for the dataset.
>
> #### **Graph pruning and compression and real-time performance prediction**
> We agree that these are very interesting directions to be explored, and we hope that our dataset will encourage more research in these areas!

---

### Official Review · Reviewer_h3jB · 2023-07-21
**Reviews to paper #659**

**Rating:** 8
**Confidence:** 3

**Strengths:**

* The paper is relevant to the broader research community as it addresses a critical issue in machine learning, which is the need for precise hardware performance models to optimize code. The dataset can be used by researchers and practitioners to develop and test new performance prediction models, which can lead to better optimization of machine learning programs.

* The paper is well-written and well-researched, with a clear focus on the contributions of the dataset. The authors provide detailed information on the dataset collection process, including the types of machine learning programs used and the compilation configurations tested. The paper also includes a thorough evaluation of the dataset, including comparisons to existing graph property prediction datasets.

Ethical and social implications are not relevant to this paper.

**Additional Feedback:**

N/A

**Clarity:**

The paper is well-written and well-researched, with a clear focus on the contributions of the dataset. The authors provide detailed information on the dataset collection process, including the types of machine learning programs used and the compilation configurations tested. The paper also includes a thorough evaluation of the dataset, including comparisons to existing graph property prediction datasets.

**Correctness:**

The dataset is constructed is solid way, and the evaluation methods are appropriate and performed correctly.

**Documentation:**

Yes, the URL of the dataset is provided, and the dataset is well documented.

**Limitations:**

* The authors need to present the limitations of the dataset, and how they can be addressed/applied on a set of different hardware architechtures. As the dataset is collected on a set of well-known architectures, it is not clear how the dataset can be applied to other architectures. The authors can consider including more machine learning programs or some testbed on different architectures.
* Beyond the dataset, the author can provide more information of how the graphs are constructured, and how the dataset can be evaluated on other HW architectures.
* Can you explain a bit of the connections to MLGO [1]? As this is the dataset, but I can see a direct integration into the compiler optimizer.

[1] MLGO: https://arxiv.org/abs/2101.04808

**Opportunities For Improvement:**

* The graph collections are preselected through a set of well-known architectures. Expanding the dataset to include more architectures can improve the generalizability of the dataset. The authors can also consider including more machine learning programs, such as those from the TensorFlow model zoo, to increase the diversity of the dataset.
* In a low level, it would be beneficial to consider a limitation of the dataset, which is the lack of information on the hardware configurations used to collect the data. The authors can consider including this information in the dataset to improve the generalizability of the dataset.


**Relation To Prior Work:**

It is well discussed how this work differs from previous contributions.

**Summary And Contributions:**

The paper introduces TPUGRAPHS, a performance prediction dataset on full tensor programs, represented as computational graphs, running on Tensor Processing Units (TPUs). The dataset contains 25x more graphs than the largest graph property prediction dataset and 770x larger graphs on average compared to existing performance prediction datasets on machine learning programs. The contribution of the paper is to provide a dataset that can assist compilers in making heuristic decisions or aid autotuners in identifying the optimal configuration for a given program. The paper primarily focuses on layout and tile configurations because tuning them offers the highest performance gain on average, compared to tuning other compiler optimizations. The layout collection is unique among existing graph datasets, in that it provides data for graph-level predictions on very large graphs.

---

> ### Author Response · Authors · 2023-08-17
>
> We thank the reviewer for the valuable feedback and insightful comments. We appreciate the reviewer for confirming that the problem we tackle is important, and our dataset is valuable.
>
> #### **Graphs in the dataset**
> We in fact include some graphs from TensorFlow model zoo (see A.3 in the Appendix), but we did not mention that in the main paper explicitly, which we will clarify. Note that not all models from the model zoo can be run on TPUs, so we would not be able to include those graphs in our dataset. However, we acknowledge that the diversity of graphs in the dataset is extremely important for the generalizability of the model, so we will definitely try to include more graphs in the next release version of the dataset if possible.
>
> Another approach to obtain more graphs is generating random graphs. However, we made a conscious decision to include only real-word graphs in the dataset because we would like a model trained on the dataset to achieve high performance on real-world graphs used in production instead of achieving moderate performance on both real-word and randomly generated graphs. In fact, we attempted to augment our dataset by *fuzzing* (mutating) the real-world graphs. One mutation was replacing nodes/operations in a graph to different operations that preserve the input/output shapes. Another mutation was changing the graph’s output tensor shape and propagating the change to the rest of the graph. Graphs generated from this process were relatively realistic because their structures are preserved. However, we did not see an improvement of the model’s quality with this data augmentation, so we did not include the augmented data in our dataset. It is possible that a more advanced graph generation that produces diverse graph structures but still similar to real-world graphs will improve the quality of the trained model. We believe this is an open-ended research question on its own. However, we acknowledge that the diversity of graphs in the dataset is extremely important for the generalizability of the model, so we will definitely try to include more graphs in the next release version of the dataset if possible.
>
> #### **Hardware configuration**
> Runtimes of all graphs are measured on a single core of TPU v3. The details on how a runtime is measured can be found in A.2 in the appendix.
>
> #### **Other hardware platforms**
> Collecting compiler configurations and runtimes for different hardware backends requires significant changes to our infrastructure, so we cannot easily construct the dataset for other hardware devices. However, we believe that the general techniques of training an accurate *learned performance model* (e.g. improvement on GNNs, Graph Segment Training method, etc) are applicable to other hardware targets, so the improvements coming out from our dataset should also benefit other hardware platforms as well. Many compiler optimizations are also common across multiple hardware backends. For example, the tile size selection has shown to be one of the most important optimizations across all widely used hardware (i.e., CPUs, GPUs, and TPUs) and even custom accelerators. Layout optimizations are also applicable on CPUs and GPUs, but the layout options on CPUs and GPUs are quite limited if the compiler depends on pre-optimized library kernels (e.g., CuDNN and MKL). We will consider including data for other hardware in future releases of the dataset.
>
> #### **Connection to MLGO**
> We can see two primary ways of using the model trained on our dataset. First, the model can be integrated with an autotuner to search for the best compiler config outside a typical compilation process. This is the main usage the paper focuses on. Second, the learned model can be used inside the compiler itself, replacing a heuristic algorithm, like in MLGO. One could use the MLGO method directly to train the model end-to-end. However, our dataset could be valuable for pretraining the model, which is then used by MLGO. Regardless, we would like to point out that MLGO is an ML framework integrated with the LLVM toolchain, so we cannot use MLGO directly in the XLA compiler because XLA doesn’t build on top of LLVM.

---

> > ### Comment · Reviewer_h3jB · 2023-08-26
> > **Response to Authors**
> >
> > Firstly, I appreciate your clarification on the inclusion of graphs from TensorFlow model zoo. I understand that not all models from the zoo can be run on TPUs, and it's great that you have made an effort to include diverse graphs in your dataset. I agree that including real-world graphs is important for the generalizability of the model, and I appreciate your decision to prioritize real-world graphs over randomly generated graphs.
> >
> > Regarding data augmentation, I understand that you have attempted to generate random graphs and mutate real-world graphs, but you did not see an improvement in the model's quality. I agree that this is an open-ended research question, and it's great that you are willing to explore it further. I suggest considering other data augmentation techniques, such as graph augmentation via node and edge insertion, deletion, and modification, which have shown promising results in other graph-related works.
> >
> > I appreciate your explanation of the hardware configuration used for measuring runtimes. It's great that you have provided detailed information on how runtimes are measured in the appendix.
> >
> > Regarding other hardware platforms, I understand that collecting compiler configurations and runtimes for different hardware backends requires significant changes to your infrastructure. However, I believe that it's important to consider other hardware platforms, especially since the techniques you are developing are applicable to other hardware targets. I suggest considering collaborations with other research groups or industry partners to collect data for other hardware platforms.
> >
> > To this end, almost all my previous concerns have been addressed. I will keep my high score as it was.

---

### Official Review · Reviewer_fGu5 · 2023-07-21

**Rating:** 5
**Confidence:** 3
**Correctness:** No major issue was identified.
**Clarity:** Clarity is good to me.

**Strengths:**

- Large-scale dataset. 25x more graphs and 770x larger graphs in terms of graph size.
- Whole graph information may be valuable for performance pass that requires a global view (e.g., layout optimization)
- Evaluation of TpuGraphs using 3-layer GraphSAGE.

**Additional Feedback:**

In lines 113-114, the authors said the XLA autotuner takes hours to converge for a single optimization pass. I am wondering if this autotuner is enabled for GPU backends. In my experience, the XLA compiler usually can compile really fast (less than 1 min or so) for common models on GPUs.

When the graph size is large, what is the prediction overhead? How does this compare to a real execution?

When talking about redundancy. Why do many samples share the same graph? If so, when splitting these samples into different sets (e.g., train, validation, test), will there be any chance of overfitting because some structures in the test set have equivalent in the training set?

**Documentation:**

The open-source link indicated in L80 https://github.com/google-research/tpu_graphs is broken (404).

**Limitations:**

My major concern is the efficiency and effectiveness when using this dataset. (1) efficiency: the computation graph of HLO can be extremely large with at least thousands of nodes. Using GNN to perform performance prediction is heavy and may need multiple accelerators for a single run (L141). Given the huge prediction overhead, how does the proposed approach compared to a real execution? Especially for the optimizations that need whole graph information such as layout optimizations. (2) effectiveness: if we categorize compiler optimization into inter-operator optimization and inner-operator optimization (e.g., how to optimize computation kernel itself). The former is the optimization that needs whole graph information as provided in this dataset. In my personal experience, inner-operator optimization contributes the majority of performance gain for most workloads. However, (as discussed in (1)) extracting features from a whole graph perspective is expensive. So my question is, does the performance improvement from the whole graph perspective outweigh the difficulty to extract a prediction from the giant whole graph? It would be nice if we can have some quantitate analysis here.

**Opportunities For Improvement:**

Please refer to the limitation section.

**Relation To Prior Work:**

The paper gives a decent discussion of previous work.

**Summary And Contributions:**

This paper proposed a large-scale dataset for tensor program performance prediction. TpuGraphs contains two collections: 1) layout collection contains the layout information of the overall graph and corresponding performance on TPUs 2) tile collection contains the fused kernel tile (typically a subset of the graph) and its performance on TPUs. The main difference between this work and previous efforts is TpuGrpahs contains the information of the whole tensor program rather than only a subgraph from the overall computation graph.

---

> ### Author Response · Authors · 2023-08-17
>
> We thank the reviewer for the valuable feedback and insightful comments. We appreciate the reviewer’s acknowledgement of the scale and importance of our dataset. We hope we are able to address the reviewer’s concerns, and respectfully ask the reviewer to consider increasing the score.
>
> #### **Efficiency of training**
> While it is true that a naive training method for our dataset, which contains very large graphs, can be inefficient and require multiple accelerators, we do provide an efficient training method (Graph Segment Training or GST for short, see Section 4.4) as a baseline for researchers to improve upon. With GST, we only need a single accelerator to train on the dataset. However, we believe that there is still a lot of room for improvements to be done for training efficiency.
>
> #### **Effectiveness of compiler optimizations targeted by the dataset**
> We believe that some people may have perceived that intra-operator (or kernel-level) optimizations contributes to the majority of performance gain for most ML workloads, because most of papers on ML compilers and optimizations focus on this type of optimizations (e,g., loop tiling, loop unrolling, vectorization, etc) since they are more tractable. However, a few recent papers [30,46,A] have shown that inter-operator (or graph-level) optimizations are extremely critical to performance as well. In our experience, we found that tile size (kernel-level) tuning and layout (graph-level) tuning offer the highest average speedups on the workloads run in our TPU fleet. Therefore, we target these two optimizations in our dataset.
>
> [30] Zhihao Jia, Oded Padon, James Thomas, Todd Warszawski, Matei Zaharia, and Alex Aiken. TASO: Optimizing Deep Learning Computation with Automatic Generation of Graph Substitutions. SOSP, 2019.
>
> [46] Phitchaya Mangpo Phothilimthana et al. A Flexible Approach to Autotuning Multi-Pass Machine Learning Compilers. PACT, 2021.
>
> [A] Wookeun Jung, Thanh Tuan Dao, and Jaejin Lee. DeepCuts: a deep learning optimization framework for versatile GPU workloads. PLDI, 2021.
>
>
> #### **Prediction overhead vs compilation and execution time**
>
> Whether or not it is worth having a model predicting runtime for graph-level optimizations depends on the specific workload: e.g., the performance gap between the optimal config and the default config (from the compiler’s heuristic), and the time (chip hours) spent searching for the best config vs the time the program runs (can be extremely large for production workloads). However, what we do know is that layout autotuning [46] discovered significant performance improvement on many of our production workloads. Without a prediction model, we have to evaluate each candidate config using compilation and execution on real hardware, which is much slower than one model prediction. Having a good prediction model will help cut down the search time significantly or may lead to even more optimal configuration within the given search time.
>
> The table below reports the evaluation time of a config when using the real evaluation (compile and execute on real hardware) and using the model prediction on graphs in the validation set. The real evaluation takes 12–3200x longer than the model prediction. This confirms that it is significantly cheaper to run a learned cost model to estimate the execution runtime, than to measure the actual runtime (which would require compilation). Also note that the graph feature extraction time and graph partition time can be amortized across multiple configs of the same graph, so the model prediction will be even faster in practice.
>
> | |&#124; Real Evaluation | Time (s) | &#124;  Model   | Prediction | Time (s)  |
> |---:|:---|---|---|---|---|
> | **Program**  | &#124; **Compilation** | **Execution** | &#124; **Feature Extraction** | **Partition** | **Inference**
> | inception_v3_batch_128_train | &#124; 135 | 3.6 |&#124;  0.3 | 0.2 | 0.1
> | mlperf_bert_batch_24_2x2     | &#124; 104 | 22 | &#124;  0.6 | 1.3 | 0.3
> | resnet_v1_50_official_batch_128_bf16 |&#124; 99 | 1.7 | &#124; 0.09 | 0.2 | 0.07
> | tf2_bert_pretrain_dynamic_batch_size |&#124; 44 | 2.8 | &#124; 0.4 | 3.3 | 0.06
> | bert_pretraining.4x4.fp16 | &#124; 45 | 2.8  | &#124; 0.2 | 3.4 | 0.3
> | resnet50.4x4.fp16         | &#124; 126 | 0.9 | &#124; 0.2 | 0.1 | 0.07
> | unet_3d.4x4.bf16          | &#124; 475 | 1.4 | &#124; 0.1 | 0.01 | 0.04
>
> XLA compilation for GPU backends is typically faster than compilation for TPU backends because (1) XLA performs minimal graph-level optimizations for GPUs, and (2) XLA mostly leverages pre-optimized CUDA kernels (e.g., CuDNN) instead of performing its down code generation.

---

> > ### Author Response · Authors · 2023-08-17
> >
> > #### **Dataset split**
> > One sample is a runtime of a graph with a specific config, so multiple samples share the same graph because of different configs. A unique graph, however, appears in only one dataset split, meaning that a graph in the test set does not appear in the training or validation set. This is because the split is done based on graphs, not samples; 10% of graphs (not 10% of samples) are chosen to be in the test set.
> >
> > #### **Incorrect dataset URL**
> > We apologize that the dataset URL provided in the main paper is incorrect. The dataset approval process — which had not been finalized when we submitted the main paper — enforces us to follow a certain naming convention, so the correct URL is https://github.com/google-research-datasets/tpu_graphs, as specified on the submission page and in the appendix.

---

### Official Review · Reviewer_cC99 · 2023-07-21
**Interesting dataset for performance prediction for tensor graphs**

**Rating:** 7
**Confidence:** 3
**Correctness:** The paper does not have any correctne…
**Clarity:** The paper is clear, well written and …

**Strengths:**

- The problem addressed in the paper is an important problem. First, building performance models for deep learning computation graphs is crucial for the efficient optimization of deep learning models. Second, releasing a large dataset for performance prediction is helpful to the research community since generating such datasets is time consuming and requires large amount of resources.
- The contribution of the paper is in releasing a dataset that supports a large computation graph, larger than graphs present in prior work.

**Additional Feedback:**

66: large-sclae --> large-scale

**Documentation:**

The paper provides many details on how the dataset was created. The dataset and code used to create it are open-source which allows reproduction. It would be useful to provide more details on the exact feature representation in the appendix though. The current description is a bit high level and leaves many low level details undiscussed.

**Ethics:**

I do not have any ethical concerns about the paper.

**Limitations:**

- Can the authors add a section to discuss their design choice to start from open-source models instead of randomly generated graphs? In particular it would be good to discuss the advantages/disadvantages of each approach and its implications.
- The authors say "To convert an unbounded list of numbers (e.g. tensor shape) to a fixed-size vector, we truncate the list to six elements and include the summation and/or product of all elements in the list", can the authors provide more insight on why this approximation is a good choice? Is it because having more than 6 tensor dimensions in the area of deep learning is rare? Or because of another reason?
- Adding a related work section would be useful. The goal is to discuss and compare prior work in a centralized section (the authors discuss prior work in the paper but they only provide a short discussion and such discussion is distributed across many sections of the paper, not centralized in a single section). Although adding a related work section is not necessary, I think it would add value to the paper.

**Opportunities For Improvement:**

Generality of the model/dataset: the dataset uses a small number of core graphs (around 7000) and then generates a large number of configurations for each graph. This might limit the ability of the model to generalize to unseen models (if they are not similar to models used in the dataset). To avoid this problem, some of the prior work, such as the Halide, Tiramisu and TVM autoschedulers/datasets generated random programs to construct the dataset. My goal is not to say that this starting for 7000 open-source models is not a good choice. I fully understand the design choice of the authors, they prefer to focus on real-world data that their model is likely to encounter in production. My point is that this design choice is important and therefore is worth discussing in the paper and even worth evaluating (evaluating the proposed performance model on computation graphs that have characteristics that are significantly different from the training dataset).

**Relation To Prior Work:**

The authors discuss prior work in many sections of the paper. It would be good to add a section devoted to prior work.

**Summary And Contributions:**

The authors introduce a new dataset for performance prediction for large tensor computation graphs run on TPUs. The dataset comprises about 40 million data points. Each data point is an optimized tensor computation graph along with its execution time on TPU. The main novelty of this dataset is that it covers larger computation graphs compared to existing datasets which cover small computation graphs. The authors provide a clear motivation on why this dataset is important, why is the problem of performance prediction challenging. They provide details on how the dataset was built, and provide a baseline model for performance prediction that achieves reasonable performance. They also provide an evaluation of the proposed baseline model.

---

> ### Author Response · Authors · 2023-08-17
>
> We thank the reviewer for the valuable feedback and insightful comments. We appreciate the reviewer for confirming that the problem we tackle is important, and our dataset is valuable.
>
> #### **Graphs in the dataset**
> The reviewer has a great suggestion on adding a discussion on our design choice on graphs included in the dataset, and we will incorporate that in the final version of the paper. As the reviewer understands, we would like a model trained on the dataset to achieve high performance on real-world graphs used in production instead of achieving moderate performance on both real-word and randomly generated graphs. In fact, we attempted to augment our dataset by *fuzzing* (mutating) the real-world graphs. One mutation was replacing nodes/operations in a graph to different operations that preserve the input/output shapes. Another mutation was changing the graph’s output tensor shape and propagating the change to the rest of the graph. Graphs generated from this process were relatively realistic because their structures are preserved. However, we did not see an improvement of the model’s quality with this data augmentation, so we did not include the augmented data in our dataset. It is possible that a more advanced graph generation that produces diverse graph structures but still similar to real-world graphs will improve the quality of the trained model. We believe this is an open-ended research question on its own. However, we acknowledge that the diversity of graphs in the dataset is extremely important for the generalizability of the model, so we will definitely try to include more graphs in the next release version of the dataset if possible.
>
> #### **Truncation to 6 dimensions**
> All the tensors appearing in the graphs in our dataset do not contain more than 6 dimensions. However, there is no guarantee that future graphs will adhere to this property, so we additionally include features containing the sum and the product of the dimension sizes, so information on the higher dimensions are not completely lost.
>
> #### **Related work**
> Section 3.2 consolidates and compares all of the related datasets against ours. It is not at a typical location (not at the beginning or the end of the paper) for related work, so it may be difficult to find. We will also consider consolidating other related work apart from related datasets in a dedicated section.
>
> #### **Input features**
> We will definitely include the description of the input features in the appendix.
>
> The following describe each element at a particular index in the node feature vector:
> ```
> 0: is_root - whether this node is the output
> 1: element_size_in_bits - deprecated, always 0
> 2–20: one hot vector of shape_element_type.
> 21–28: size (number of elements) for each dimension, or an upper bound on the size if the dimension is dynamic.  In XLA, dimensions are numbered from 0 to N-1 for an N-dimensional array. The first element of 'shape_dimensions' is the size of dimension 0, the second element is the size of dimension 1, and so forth.  Empty list indicates a scalar.
> 29: shape_tuple_shapes_size - for tuples only, the shapes of constituent shapes in the tuple sequence.
> 30: parameter_number = K - indicating that is is the Kth parameter to the computation, only for Parameter operation
> 31–36: dimensions present for some operations that require reshaping or broadcasting, including Reshape, Reduce, ReduceWindow, and Reverse.
> 37–92: windowing information in an operation such as convolution. The window is moved across a base area and for each position of the window a computation is performed.
> 93–106: dimension numbers used for a convolution.
> 107: feature_group_count - the number of feature groups, used for a convolution. Must be a divisor of the input feature dimension and output feature dimension. If not specified, it will use a default value of 1.
> 108: batch_group_count - the number of batch groups, used for a convolution.
> 109–120: [begin/start, end/limit) index range and stride for a slice operation.
> 121 - 124: [start, start + size) range size for a dynamic slice ('start' is specified dynamically in the second operand of the operation).
> 125–132: padding configuration that describes the edge padding of a pad operation.
> 133: is_stable - whether this Sort operation should be stable
> 134–139: physical layout used to pack the tensor shape.
> ```
>
> The following describe each element at a particular index in the tile config feature vector.
> ```
> 0–5: tile sizes of the convolution kernel, only for a convolution operation.
> 6–11: output tile sizes.
> 12-17: iutput tile sizes.
> ```
>
> The following describe each element at a particular index in the per-node layout config feature vector.
> ```
> 0–5: physical layout of the output tensor
> 6-11: physical layout of the input  tensor
> 12-17: physical layout of the kernel tensor, only for a convolution operation
> ```

---

### Decision · Program_Chairs · 2023-09-22

**Decision:**

Accept (Poster)

**Comment:**

In summary, the paper provides realistic data as put by one of the reviewers a  "large-scale dataset. 25x more graphs and 770x larger graphs in terms of graph size" as well a performance model that may serve future optimizations attempts in deep learning.

The authors responded to many of the requests and criticism of the reviews to improve their submission which was acknowledged by some of the reviewers.

Despite the difficulty in running the approach on different hardware platforms, we'd still like to encourage the authors to pursue this endeavor to make the work even more useful and impactful.